# Centralization of Major Trauma Influences Liver Availability for Transplantation in Northern Italy: Lesson Learned from COVID-19 Pandemic

**DOI:** 10.3390/jcm11133658

**Published:** 2022-06-24

**Authors:** Michele Altomare, Andrea Chierici, Francesco Virdis, Andrea Spota, Stefano Piero Bernardo Cioffi, Shir Sara Bekhor, Luca Del Prete, Elisa Reitano, Marco Sacchi, Federico Ambrogi, Osvaldo Chiara, Stefania Cimbanassi

**Affiliations:** 1Department of Surgical Sciences, Sapienza University of Rome, Piazzale Aldo Moro 5, 00185 Rome, Italy; 2General Surgery and Trauma Team, ASST Niguarda, Milano, Piazza Ospedale Maggiore 3, 20162 Milan, Italy; francesco.virdis@ospedaleniguarda.it (F.V.); andrea.spota@ospedaleniguarda.it (A.S.); stefanopiero.cioffi@ospedaleniguarda.it (S.P.B.C.); shir.bekhor@studenti.unimi.it (S.S.B.); osvaldo.chiara@unimi.it (O.C.); stefania.cimbanassi@unimi.it (S.C.); 3Centre Hospitalier d’Antibes Juan-les-Pins–Chirugie Digestive, Department of General and Emergency Surgery, 06600 Antibes, France; andrea.chierici@ch-antibes.fr; 4General and Liver Transplant Surgery Unit, Fondazione IRCCS Ca’ Granda Ospedale Maggiore Policlinico, 20122 Milan, Italy; luca.delprete@unimi.it; 5General and Emergency Surgery, Ospedale Maggiore Della Carità di Novara, 28100 Novara, Italy; elisa.reitano@live.it; 6Department Emergenza Urgenza-E.A.S. SOREU Metropolitana, 20161 Milan, Italy; marco.sacchi@ospedaleniguarda.it; 7Department of Clinical Science and Community Health, University of Milan, Festa del Perdono 7, 20122 Milan, Italy; federico.ambrogi@unimi.it; 8Department of Medical-Surgical Physiopathology and Transplantation, University of Milan, Festa del Perdono 7, 20122 Milan, Italy

**Keywords:** Trauma, Damage Control Strategy, organ donation, trauma donors, liver donation, liver transplant

## Abstract

Background: During the COVID-19 pandemic, the centralization of patients allowed trauma and transplants referral centers to continue their routine activity, ensuring the best access to health care. This study aims to analyze how the centralization of trauma is linked with liver allocation in Northern Italy. Methods: Cluster analysis was performed to generate patient phenotype according to trauma-related variables. Comparison between clusters was performed to evaluate differences in damage control strategy procedures (DCS) performed and the 30-day graft dysfunction. Results: During the pandemic period, the centralization of major trauma has deeply impaired the liver procurement and allocation between the transplant centers in the metropolitan area of Milan (Niguarda: 22 liver procurement; other transplant centers: 2 organ procurement). Two clusters were identified the in Niguarda’s series: cluster 1 is represented by 17 (27.4%) trauma donors, of which 13 (76.5%) were treated with DCS procedures, and 4 (23.5%) did not; cluster 2 is represented by 45 trauma donors (72.6%), of which 22 (48.8%) underwent DCS procedures. A significant difference was found in the number of DCS procedures performed between clusters (3.18 ± 2.255 vs. 1.11 ± 1.05, *p* = 0.0001). Comparative analysis did not significantly differ in the number of transplanted livers (cluster1/cluster2 94.1%/95.6% *p* = 0.84) and the 30-day graft dysfunction rate (cluster1/cluster2 0.0%/4.8% *p* = 0.34). Conclusions: The high level of care guaranteed by first-level trauma centers could reduce the loss of organs suitable for donation, maintaining the good outcomes of transplanted ones, even in case of multiple organ injuries. The pandemic period underlined that the centralization of major trauma impairs the liver allocation between transplant centers.

## 1. Introduction

The COVID-19 pandemic has had a massive impact on healthcare systems, affecting all aspects of medical care. Several studies reported substantial organ donation and transplantation decreases worldwide due to the ongoing pandemic [1,2,3]. This global reduction of transplant activity was noted especially among centers where COVID-19 had a higher incidence, and thus faced an associated depletion of intensive care unit (ICU) beds [4,5]. In addition, the number of potential donors decreased as the rate of neurocritical patient admissions lowered. Organs from potential donors with a positive or unknown SARS-CoV-2 status were discarded, further contributing to the reduction in deceased organ donation [6,7]. With regards to liver transplantation, it was evident that transplant teams faced significant difficulties in allocating a life-saving procedure during the pandemic. Literature on liver transplantation during COVID-19 pandemic is scarce, with unclear data collected in burdensome conditions [8]. Liver procurement from trauma patients has also suffered the consequences of the pandemic. The adopted preventive measures, including stay-at-home orders, travel restrictions, and smart working, led to a drastic fall in the number of acute injuries. Due to hospitals’ saturation, emergency measures included centralizing patients to referral centers [9]. The hub and spoke model was adopted from the emergency setting and applied to different fields of surgery (e.g., oncologic) [10,11]. This centralization allowed referral centers to continue their routine activity, thus ensuring the best access to health care for non-COVID patients [12]. Following this principle, first-level trauma centers were overwhelmed by trauma admissions during the pandemic, even when a general decrease in traumatic events was registered. To better understand how this centralization of trauma patients could have affected organ allocation in Italy, it is essential to describe the two main mechanisms which guide organ donation in Italy: first, every transplant group has its own ICU which provides internal donation; second, the allocation follows a chronological list based on patients’ priority and is regulated by a national transplant center (CNT) and a regional center of reference (CRR–NITp) for Northern Italy only [13,14]. During the pandemic, only major trauma centers were identified as hubs for traumatic injury, pointing out the centralization of potential donors. 

The present study aims to analyze how the centralization strategy for trauma and acute care surgery in Northern Italy affects liver donation and allocation, focusing on the period of the COVID-19 pandemic (2020–2021). 

## 2. Materials and Methods

The present study is a spin-off project of an ongoing multicentric retrospective study collecting data on trauma donors (TD) from international trauma and solid organ transplantation referral centers. Data reported include preliminary results from ASST GOM Niguarda, the coordinating center of this project. Other unpublished results on the complete cohort will be presented at the next European Congress of Trauma and Emergency Surgery (ECTES–2022). A retrospective, observational cohort study was conducted through a complete data review of consecutive trauma patients who underwent organ donation (OD) between January 2012 and January 2022. Inclusion criteria were the following: at least one procured and transplanted organ; availability of all records regarding emergency department (ED) admission and Damage Control Strategy (DCS) management; availability of data regarding organ donation and short-term functional outcome of transplanted organs. Cluster analysis was performed to generate patient phenotype according to trauma-related variables. The partitioning around medoids (PAM) algorithm using Gower’s distance was the selected approach, using the following variables collected throughout the study: Sex; Age; BMI; prehospital systolic blood pressure (SBP); prehospital heart rate (HR); prehospital Glasgow Coma Scale (GCS); prehospital cardiac arrest (CA); prehospital Tranexamic acid (TXA); prehospital colloids or crystalloids; prehospital endotracheal intubation (IOT); prehospital bilateral decompressive thoracotomies; prehospital vasoactive usage; CA at admission; SBP at entrance; HR at entry; GCS at admission; shock index at admission; ABG data at admission (Lactate level, Base excess, pH); total transfusion during hospital stay; Damage control surgery procedures performed in emergency department (ED); thoracostomy in ED; extraperitoneal pelvic packing (EPP) in ED; emergency department thoracotomy (EDT); ED-TXA; ED vasoactive usage; massive transfusion protocol (MTP) activation in ED; injury severity scale (ISS); presence of solid organ injuries (Brain, Liver, Kidney, Lung, Heart). 

Clusters were compared according to the percentage of patients undergoing DCS. To determine the optimal number of clusters, the silhouette method was used. The DCS procedures included in the dataset were the following: monolateral or bilateral decompressive thoracostomies; extraperitoneal pelvic packing (EPP); EDT; REBOA^®^ positioning in the emergency department or operative room; MTP activation; exploratory and/or decompressive laparotomy; exploratory thoracotomy; decompressive craniotomy; external bone fixation; therapeutic angiography with embolization. The DCS-group is defined as at least one DCS procedures performed. Mann–Whitney–Wilcoxon and the Fisher–Freeman–Halton tests were used to analyze the association between categorical variables. Results with *p* < 0.05 were considered statistically significant. Short-term outcomes of transplanted organs were evaluated in terms of 30-day graft dysfunction requiring explanation. All subjects included in the analysis underwent donation after brain death (DBD). Nowadays, in Italy, the pool of donations after cardiac death (DCD) includes mostly type III DCD and non-trauma-related. Cardiac arrest in trauma patients often leads to the onset of the *triad of death* (acidosis, coagulopathy, and hypothermia) despite highly aggressive management. This condition hardly maintains the physiological reserve of injured organs. Liver injuries were classified using the Abbreviated Injury Scale 2015 revision. All organ procurements were conducted following the standard surgical procedures. 

To better understand the results of our study population and the distribution of liver donation in the metropolitan area of Milan, the number of liver donations in the period of 2020–2021 was retrieved from two other transplant centers in the city; Policlinico of Milano and National Cancer Institute (Istituto Nazionale Tumori, INT). As mentioned, ASST–GOM Niguarda is a Level I Trauma Center. On the contrary, Policlinico of Milan is a Level II Trauma Center. The INT doesn’t have an Emergency Department but is affiliated to the ICU of the Hospital of Cremona, a Level IV trauma center. During the first wave of the pandemic, the emergency department of ASST-GOM Niguarda was the referral center also for cerebrovascular accidents. Data were collected in a computerized spreadsheet (Microsoft Excel 2016; Microsoft Corporation, Redmond; WA) and analyzed with statistical software (IBM SPSS Statistics for Windows, version 25.0, IBM Corp., Armonk, NY, USA).

## 3. Results

### 3.1. Study Population

Sixty-four patients underwent organ procurement following trauma during the study period at ASST GOM Niguarda. Two patients (3.2%) were excluded from the initial cohort due to missing data regarding emergency department admission and organ donation. Thus, sixty-two (96.9%) patients were included in the final group. The mean age was 48.9 (95%CI 43.8–54.2). Forty-two donors were male (67.7%), and 20 (32.3%) were female, with a male/female ratio of 2.1. The Charlson Comorbidity Index (CCI) distribution is shown in Figure 1. The mean Body Mass Index (BMI) is 24.8 (95%CI 23.9–25.9). The mean Shock Index was 0.95 (95%CI 0.80–1.01). Fifty-eight patients (93.5%) had blunt trauma, and the remaining four (6.5%) cases were penetrating. The mean ISS and new ISS (NISS) were respectively 43.9 (95%CI 39.8–48.0) and 52.2 (95%CI 48.9–55.5). Other relevant clinical parameters, demographic characteristics, pre-hospital rescue, and ED shock room (SR) parameters are summarized in Table 1. 

Almost one-third (35.4%) of the study population (i.e., 22 trauma donors) were evaluated during the pandemic period. The distribution of liver donors during the study period is shown in Figure 2. The liver/donor ratio before and after the pandemic was respectively 0.93 and 1 (*p* > 0.05).

Figure 3 demonstrates the distribution of livers allocation during the pandemic (i.e., 2020–2021) in the transplant centers of the metropolitan area of Milan. The numbers reported are related to liver donation only and don’t represent the total number of transplants performed. Those data show how during the pandemic the centralization of major trauma has deeply affected the allocation of trauma donors between transplant groups in the metropolitan area of Milan. Our center did twenty-two organs procurement from trauma donors during the pandemic period compared to only two liver procurement performed by other centers.

### 3.2. Cluster Analysis

Two clusters were identified from the PAM algorithms and are shown in Figure 4. Each donor is marked with *Yes* or *No* according to whether they were managed with an aggressive damage control strategy (DCS) approach or not. Cluster 1 is represented by 17 (27.4%) trauma donors, of which 13 (76.5%) were treated with DCS procedures, and 4 (23.5%) did not. Cluster 2 is represented by 45 trauma donors (72.6%), of which 22 (48.8%) underwent DCS procedures. Comparison between the clusters discloses that patients in Cluster 1 had a worse clinical condition both in the prehospital setting and in the ED. Pre-hospital and in-hospital principal variables are summarized in Table 2, and complete results of the comparative analysis are reported in Appendix A Table A1. Even though no significant difference between groups was found regarding the overall need for DCS approach (76.5% vs. 48.9%, *p* = 0.083), a significant difference was found in the number of DCS procedures performed (3.18 ± 2.255 vs. 1.11 ± 1.05, *p* = 0.0001). Nevertheless, no significant differences were registered in the total number of procured livers (16(94.1%) vs. 43(95.6%) *p* ≥ 0.05).

### 3.3. Outcomes of the Transplanted Organs

Fifty-nine (95.1%) livers were procured with 6 (10.3%) split donor procurement. Thus, a total of 65 organs were transplanted with a liver/donor ratio of 1.04. Data concerning 30-day organ dysfunction was available for only 34 (54.6%) of the transplanted livers, of which only one case had primary non function which required explantation, thus functional response rate was 97%. Comparative analysis between the clusters did not significantly differ in the number of procured livers and the 30-day graft dysfunction rate. Results are reported in Table 3. Of 59 procured livers, four (6.7%) were transplanted although suffered a trauma-related injury. No graft dysfunction or short-term complications were registered in this group. The first of which had an AIS 3 injury of the right lobe. A 5 cm parenchymal depth laceration without major duct involvement and a sub-capsular non-expanding intraparenchymal hematoma of 5 cm were present between segments five and six. The injury was treated conservatively without damage control laparotomy or embolization. A massive transfusion protocol was activated, and bilateral thoracotomies were performed in the scene. Other two patients had minor injuries with two AIS 2 lesions. The first injury was on the left lobe with a superficial laceration of the second segment and the second one a superficial subcapsular hematoma smaller than 10 cm. The fourth case was a 53 years-old woman with an AIS 4 injury of the right lobe due to the trauma and the wrong positioning of the cardiopulmonary resuscitating device (LUCAS II). She underwent damage control laparotomy with four-quadrant packing and subsequent angioembolization of the left branch of the hepatic artery. The full description of this peculiar clinical case with associated images is published as a case report by the transplant group that procured the liver [15].

## 4. Discussion

During the COVID-19 pandemic, starting from the beginning of year 2020, essential changes in trauma admissions have been reported. Apart from an overall decrease in trauma prevalence and admission, primarily due to the shelter-in-place guidelines and the consequent reduction of motor vehicles accidents, blunt traumas, and penetrating traumas, a nationwide retrospective study by Berg et al. [16] on American trauma center showed a tendency of concentration of trauma referrals to level I trauma centers. Lombardy region has been significantly affected by the COVID-19 pandemic. It was the first European region to experience the disease’s outbreak, and trauma patients were almost exclusively referred to Level 1 trauma centers such as ASST-GOM Niguarda Hospital. As illustrated by our results, this lead to an increase in the rate of OD from trauma patients after resuscitation attempts compared to other transplant centers in the metropolitan area of Milan. The pandemic exemplified how the centralization of major trauma could affect the availability of organs, confirming the central role of trauma surgeons as stewards of trauma donor resources. The trauma system in Italy is based on a network of referral centers. Urban areas are organized in integrated trauma systems (Sistema integrato per l’assistenza per il trauma maggiore, SIAT) which geographically provides a network of trauma hospitals categorized into three levels: high-specialization trauma center (CTS or Level 1 trauma center) with all specialists available 24/7; areal trauma center (CTZ or level 2); and emergency hospital for trauma (PST or level 3) which is located in remote areas and holds resources only for patients stabilization. The COVID-19 pandemic proved the efficacy of this already present and well-working mechanism in the management of major trauma. Although centralization has improved the outcomes of trauma patients [17], traumatic brain injury still represents a major cause of mortality [18,19]. Trauma donors are generally younger and healthier than non-trauma donors and constitute a valuable pool for organs procurement. In a recent retrospective analysis of the United States Scientific Registry of Transplant Recipients, Ackerman et al. [20] reported that trauma patients produce more transplanted organs per donor when compared to non-trauma patients and with better organs characteristics when considering kidneys specifically; this finding does not seem to be valid also for liver transplantation from trauma donors, literature is lacking in this field. In a recent monocentric cohort study, we described a cohort of trauma donors after damage control strategy from 2018 to 2021. Our experience confirms the elevated organ/patient ratio of donation from trauma patients (3.8). Moreover, focusing on liver donation, a 93.3% rate of OD was found, with 3.6% primary non-function rate, 0% 30-day mortality related to graft disfunction, and 13.3% of donors with an AIS II-IV liver injury [21]. This means that trauma-related organ damage per se should not be a reason to discard a liver, and that special attention should be given in the management of potential trauma donors to increase available organs. For this reason, research of eventual factors that may influence the possibility of donation and outcomes of transplantation is of paramount importance. Our study denotes that direct and indirect factors associated with severe trauma, and thus increased risk of multi-organ failure (lower pre-hospital GCS, pre-hospital cardiac arrest, increased Shock Index, and lower BE in the ED) do not seem to affect OD and the functional response rate of the transplanted liver. Alarhayem et al. [16,22] published a retrospective analysis of outcomes of patients “dead on arrival” who underwent aggressive resuscitation reporting a 3.6% rate of organ donation; 25% of them were livers. Although the rate is meager, aggressive resuscitation should always be performed as it can lead to valuable benefits such as OD, which is clearly sustained by our results. Moreover, considering the specificity of each transplant center, even if the centralization of potential donors increases the survival chance of procured organs, the careful allocation process guaranteed by the CNT and CRR avoids the risk of unregulated allocation among different transplant centers. The results of this study underlie a possible criticism of the actual Italian allocation mechanism for organ donation which was pointed out thanks to the pandemic.

To the best of our knowledge, this is the first manuscript focusing on the definition of factors that could affect liver donation from trauma donors with an insight into trauma centralization and the recent pandemic experience. Furthermore, we used machine learning to perform clustering to obtain two different peculiar subpopulations to be compared to search for risk factors associated with organ donation; the advantage of PAM is that it makes the analysis less sensitive to outliers.

There are several limitations to this study deriving from its retrospective nature and the use of electronic medical records. The restricted number of patients included in the cohort represents the principal drawbacks of this study. Trauma management is highly dynamic and the interpretation of the events depends on the accuracy of their description and the level of details from both pre-hospital and in-hospital patient records. We acknowledge that our definition of functional response considering 30-days post-transplant is relatively short-term and was chosen due to the attainability of data.

## 5. Conclusions

The number of liver trauma donors in our center is highly increased during the COVID-19 pandemic. Moreover, the high level of care guaranteed by first-level trauma centers could reduce the loss of organs suitable for donation, maintaining the good final outcomes of transplanted ones, even in case of organ injury. The pandemic period underlined that the centralization of major trauma increases the availability of organs for transplantation in a single center: this factor needs to be taken into consideration reasoning in further allocation strategies at the regional and national levels.

## Figures and Tables

**Figure 1 jcm-11-03658-f001:**
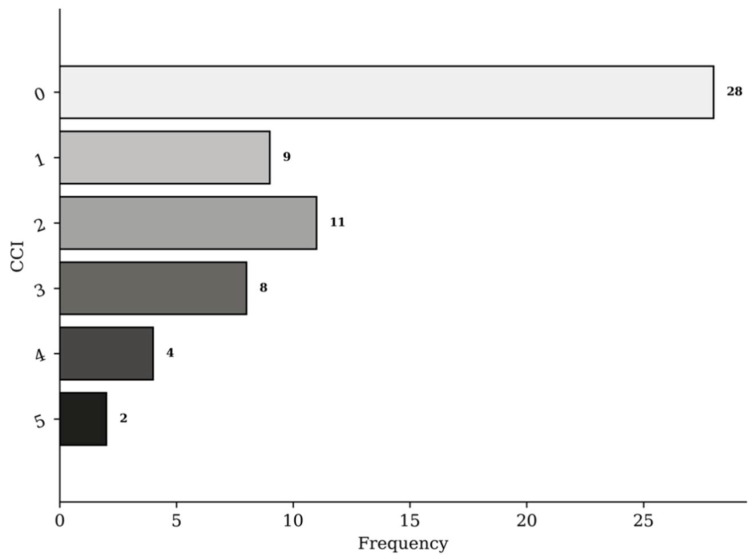
Charlson Comorbidity Index (CCI).

**Figure 2 jcm-11-03658-f002:**
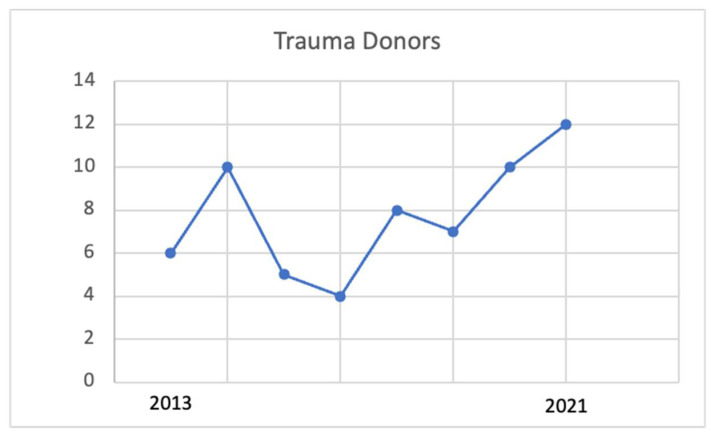
Distribution among years of traumatic liver donors at ASST GOM Niguarda.

**Figure 3 jcm-11-03658-f003:**
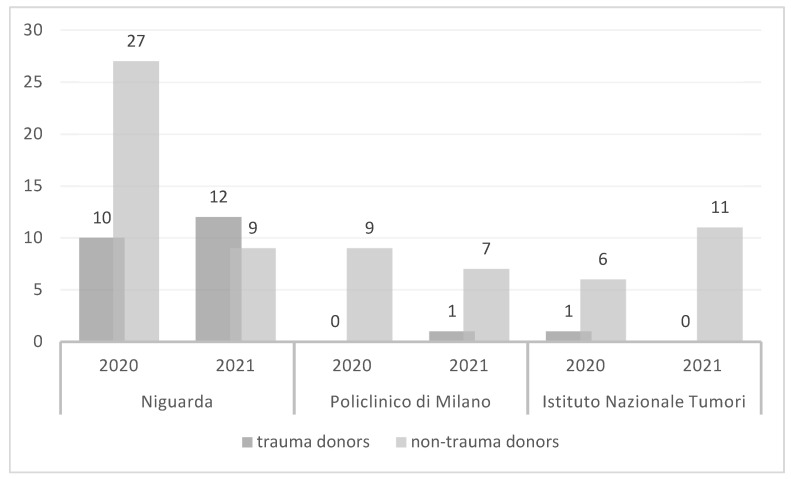
Organs allocation in the metropolitan area of Milan during the pandemic (i.e., 2020–2021). Histograms show Liver donation per center according to donor type (trauma vs. non-trauma). The ASST–GOM Niguarda is a Level I Trauma Center. Policlinico of Milan is a Level II Trauma Center. INT doesn’t have an Emergency Department but is affiliated with the ICU of the Hospital of Cremona, a Level IV trauma center. As already mentioned in the materials and methods section, it is essential to underline that during the first wave of the pandemic, the emergency department of ASST-GOM Niguarda was the referral center also for cerebrovascular accidents, which explain the high number of non-trauma related donors.

**Figure 4 jcm-11-03658-f004:**
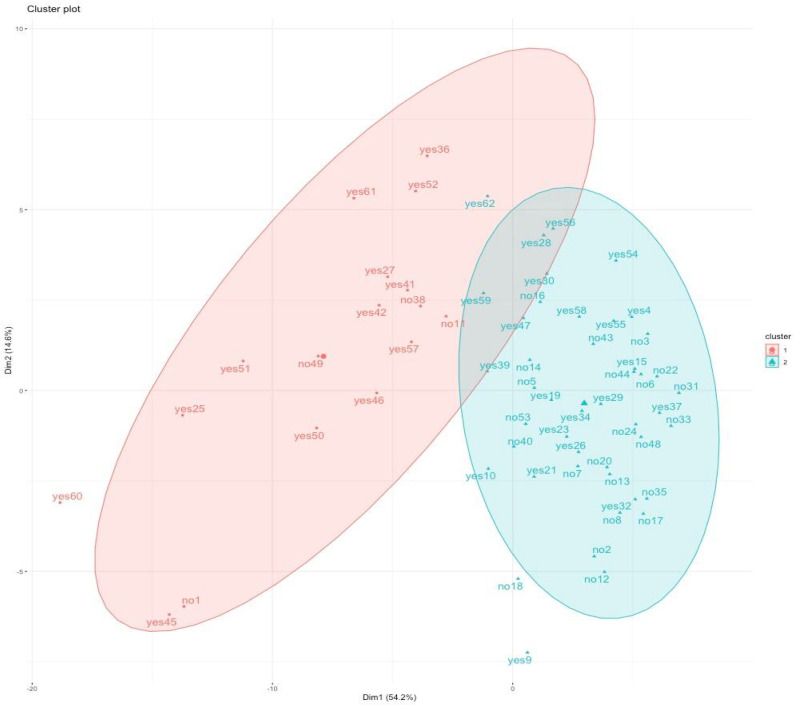
Cluster analysis of trauma-related variables in DBD. Each number matches one donor, and marked with *Yes* or *No* according to whether they were managed with DCS or not. The analysis reaches to divide the population into two clusters according to the variables described in the materials and methods section.

**Table 1 jcm-11-03658-t001:** Relevant parameters during the prehospital and in-hospital evaluation.

	Mean	Median	95% Confidence Interval
ED–SBP	106.4	120	[92.2–120.7]
ED–HR	87.6	90	[76.9–98.3]
ED–GCS	4.75	3	[4.0–5.4]
pH	7.17	7.30	[7.12–7.22]
BE	−8.26	-6.95	[−10.1–−6.3]
Lactate Level	5.31	3.9	[4.3–6.3]
Total Transfusion	25.7	21.5	[20.1–31.2]
Number of DCS procedures	1.68	1	[1.2–2.1]

SBP: Systolic Blood Pressure; HR: heart rate; GCS: Glasgow Coma Scale; ED: emergency department; BE: base excess; DCS: Damage Control Strategy procedures.

**Table 2 jcm-11-03658-t002:** Variables related to the clinical condition of major trauma patients in the prehospital setting and Emergency Department (ED). Differences between clusters. The complete results of this analysis are shown in Appendix A.

	Cluster 1 (17)	Cluster 2 (45)	*p*-Value
Pre-hospital GCS	3.29 ± 0.749	5.32 ± 3.117	*p* = 0.0073 **^,a^
	(range: 3–6)	(range: 3–15)	
ED-Shock Index	1.40 ± 0.840	0.78 ± 0.352	*p* = 0.0013 **^,a^
	(range: 0–3.67)	(range: 0–2)	
ED-Base Excess	−15.06 ± 7.526	−5.70 ± 5.985	*p* = 0 ****^,a^
	(range: −30–1.6)	(range: −25–5.8)	
Pre-hospital cardiac arrest	Yes 10 (58.8%)	Yes 4 (9.1%)	*p* = 0.00012 ***^,b^
	No 7 (41.2%)	No 40 (90.9%)	
NISS	58.29 ± 15.430	49.91 ± 11.623	*p* = 0.0232 *^,a^
	(range: 20–75)	(range: 29–75)	

* *p* < 0.05; ** *p* < 0.01; *** *p* < 0.001; **** *p* < 0.0001; ^a^ Mann-Whitney non-parametric test for unpaired samples; ^b^ 2-tailed Fisher’s exact test for categorical data.

**Table 3 jcm-11-03658-t003:** Outcomes of transplanted liver from trauma donors (TDs).

	Cluster 1 (17)	Cluster 2 (45)	*p*-Value
N° of donated liver	Yes 16 (94.1%)	Yes 43 (95.6%)	*p* = 1 ^b^
	No 1 (5.9%)	No 2 (4.4%)	
N° of donated hemi-liver	Yes 2 (11.8%)	Yes 4 (8.9%)	*p* = 0.66 ^b^
	No 15 (88.2%)	No 41 (91.1%)	
N° of transplanted liver	Yes 16 (94.1%)	Yes 43 (95.6%)	*p* = 0.84 ^b^
	No 1 (5.9%)	No 2 (4.4%)	
	**Cluster 1 (13)**	**Cluster 2 (21)**	** *p* ** **-Value**
Liver functional response	Yes 13 (100%)	Yes 20 (95.2%)	*p* = 0.34 ^b^
	No 0 (0%)	No 1 (4.8%)	

^b^ 2-tailed Fisher’s exact test for categorical data.

## Data Availability

The data presented in this study are available on request from the corresponding author. The data are not publicly available to preserve confidentiality.

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
