# Peer review of "Centralization of Major Trauma Influences Liver Availability for Transplantation in Northern Italy: Lesson Learned from COVID-19 Pandemic"

_jcm, 2022, doi:10.3390/jcm11133658_

Round 1
Reviewer 1 Report
Altomare et al. aimed to analyze how the centralization of trauma is linked with the liver allocation in Northern Italy: Cluster analysis was performed to generate patient phenotype according to trauma-related variables. Comparison between clusters was performed to evaluate differences in damage control strategy procedures (DCS) performed and the 30-day graft dysfunction. According to the study a significant difference was found in the number of DCS procedures performed between clusters ().They concluded as the high level of care guaranteed by first-level trauma centers could reduce the loss of organs suitable for donation, maintaining the good final outcomes of transplanted ones, even in case of multiple organ injury. They reported as the pandemic period underlined that the centralization of major trauma impairs the liver allocation between transplant centers.The idea is smart. Some minor issues raised;
1-Page 9, line 274, please delete point before sentence.
2-There are some similar (not the same) studies in the current medical literature
For example:Changes in the Deceased-Donor Trend in Korea: Establishment of Regional Trauma Centers and KODA.Lee JM.J Clin Med. 2022 Feb 24;11(5):1239. doi: 10.3390/jcm11051239.
Would you like to discuss these studies?
3-Please add a limitation paragraph before the conclusion. So, we can see the strengths and weakness of the study.
Thank you giving opportunity to review this study.
Author Response
We would also like to thank the Reviewers for their valid and accurate suggestions, which helped us improve the general quality of the description. According to the Reviewer's requests, we made the following adjustments to the revised version R1 of our article.
Reviewer 1:
- Page 9, line 274, please delete the point before the sentence.
We have deleted the point before the paragraph.
1.2 There are some similar (not the same) studies in the current medical literature. For example, Changes in the Deceased-Donor Trend in Korea: Establishment of Regional Trauma Centers and KODA.Lee JM.J Clin Med. 2022 Feb 24;11(5):1239. DOI: 10.3390/jcm11051239.
Would you like to discuss these studies?
We really would like to thank reviewer 1 for the suggested reading. The article from Lee et al. analyzes how the trend of deceased-donor recruitment and donation has changed in South Korea based on policy factors such as independent organ-procurement organization (IOPO) activities and establishing regional trauma centers. Despite the similarity of the contents in the two articles, there are profound differences between each country regarding the organs allocation, and comparing the population could be misleading. We are planning to build a national trauma donors registry to study this population's Italian epidemiology, and we could utilize the suggested article in the following paper.
- Please add a limitation paragraph before the conclusion. So, we can see the strengths and weaknesses of the study.
As requested, this R1 enclosed version adds a new limitation paragraph (line 307-313) before the conclusion.
To satisfy the minor English revisions requested from Reviewer 2 and Reviewer 3, thanks to a mother tongue colleague, we also extensively edited the English language and style.
We declare that the article is original and was never presented and/or submitted elsewhere, and all Authors approve the revised version.
We hope that the enclosed amended R1 version of the present submission will be judged of sufficient quality for being accepted for publication in JCM.
We remain at your disposal if any question or request may arise.
Respectfully yours,
Michele Altomare, MD
Sapienza University of Rome, Rome (Italy)
michele.altomare@uniroma1.it
Reviewer 2 Report
Dear authors
Congratulations on writing this interesting paper. The actual impact of COVID outbreak on organ transplantation has been seen across the board and the system adopted to allocate severe trauma to specialised centres address the point of saving potential organ donors.
Some minor spelling checks and typos only.
Regards
Author Response
Please see the file attached.

Reviewer 3 Report
Thank you for addressing the minor concerns. Congratulations on a good paper
Author Response
Please see the file attached.

This manuscript is a resubmission of an earlier submission. The following is a list of the peer review reports and author responses from that submission.
Round 1
Reviewer 1 Report
Congratulations on a high percentage of successful transplants from a sick population of DBD trauma donors. Centralization of trauma referral and aggressive management of sick (Cluster 1) patients with DCS procedures has resulted in outcomes comparable to slightly less sicker (Cluster 2) patients
For the benefit, interest of readers could you describe in a little more detail how these Clusters were derived.
In the discussion the difference in the type of DCS procedures performed between the two Clusters could be highlighted
Minor points
In the abstract should the 30 day graft dysfunction rate be 0. 4.8 %
Are references 3 and 4 the same?